# Comprehensive Characterization of Oils and Fats of Six Species from the Colombian Amazon Region with Industrial Potential

**DOI:** 10.3390/biom13060985

**Published:** 2023-06-13

**Authors:** Kimberly Lozano-Garzón, Luisa L. Orduz-Díaz, Camilo Guerrero-Perilla, Willian Quintero-Mendoza, Marcela P. Carrillo, Juliana E. C. Cardona-Jaramillo

**Affiliations:** 1Instituto Amazónico de Investigaciones Científicas Sinchi, Calle 20 # 5-44, Bogotá 110311, Colombia; klozano@sinchi.org.co (K.L.-G.); lorduz@sinchi.org.co (L.L.O.-D.); wiquintero@sinchi.org.co (W.Q.-M.); mcarrillo@sinchi.org.co (M.P.C.); 2Facultad de Ingeniería, Departamento de Ingeniería Química, Universidad de la Sabana, Km. 7, Autopista Norte de Bogotá, Chía 250001, Colombia; 3Facultad de Medicina y Ciencias de la Salud, Universidad Militar Nueva Granada, Km. 2, vía Cajicá-Zipaquirá, Cajicá 250247, Colombia

**Keywords:** Amazonian oils, bioactive compounds, natural ingredients, oil quality, bioproducts

## Abstract

The Colombian Amazon is a megadiverse region with high potential for commercial use in the pharmaceutical, food, and cosmetic industries, constantly expanding and looking for new alternatives from natural resources; unfortunately, few characterization reports of its profitable non-timber species in Colombia have been conducted. This work aimed to perform a comprehensive analysis of traditionally used species: *Carapa guianensis* (Andiroba), *Euterpe precatoria* (Asai), *Mauritia flexuosa* (Miriti), *Astrocaryum murumuru* (Murumuru), *Plukenetia volubilis* (Sacha Inchi), and *Caryodendron orinocense* H.Karst (Cacay). For this purpose, oil and fat quality indices, phytosterol, carotenoid, tocopherol, and tocotrienol content, as well as density and refractive index, were measured to establish their quality level. Multivariate analysis showed four groups of samples; such differences were mainly due to the composition rather than quality indices and physical properties, especially the content of saturated and unsaturated fatty acids. All species reported a precise composition, which makes them noninterchangeable, and Miriti oil arose as the most versatile ingredient for the industry. The Colombian Amazon region is a promising source of quality raw material, especially for oils/fats and unsaturated fatty acids; this resulted in the most interest for pharmaceutical, food, and cosmetic purposes.

## 1. Introduction

The natural ingredient industry can be classified into two different types of products: those of a biological and agronomic nature (source products, i.e., the species, plant, or its parts, from which the active ingredients of interest are extracted) and those of a chemical nature (use products, i.e., the ingredients themselves, extracted or derived from the source products employing chemical processes developed through technology). While the source products tend to constitute a dispersed portfolio with technological specificities specific to each species and its environment, the use products tend to be concentrated in industrial laboratories with increasingly complex technologies but less specific to each product. The pharmaceutical, food, and cosmetic (PFC), and final product industries generate a higher demand for natural ingredients (NI) [1]. In 2019, about 9.8 million tons of natural ingredients were imported worldwide, corresponding to USD 29,505 million. The most demanded raw material in 2019 was vegetable extracts, with 23.5% of the total market [2]. The demand for natural ingredient-based products is increasing worldwide. In this context, Colombia, as the second most biodiverse country worldwide [3], is in a privileged position to take advantage of this demand, enhance its social and technological development, and become a leading player in natural ingredient production for PFC industries. It is also an opportunity to consolidate value chains that take into account the environmental and social dimensions by taking advantage of biodiversity and empowering communities that have been historically isolated while at the same time protecting natural resources and generating sustainable use of biodiversity.

Based on prospecting studies and market demand for NI, the Colombian government has identified this sector as an opportunity due to the country’s vast biodiversity [4]. Consequently, Colombian society has a critical need for projects that aim to enhance the performance of the national NI sector. Harvesting natural resources, processing NI, and formulating new products will provide new employment opportunities for economic reactivation. The sector’s size is expected to quintuple (USD 15.4 billion), increase sevenfold for NI exports, and generate 46,700 direct jobs [5].

Achieving this goal involves a high degree of expertise often unavailable to small organizations in the Colombian Amazon region that aim to sell NI to the cosmetics industry. Studies of this nature help make relevant information accessible regarding the chemical composition of these biodiversity assets, which can serve as a source of oils for cosmetics and functional industries.

Species such as *Carapa guianensis* (Andiroba), *Platonia insignis* (Bacuri), *Euterpe precatoria* (Asai), *Mauritia flexuosa* (Miriti), *Oenocarpus bataua* (Patawa), *Astrocaryum murumuru* (Murumuru), *Plukenetia volubilis* (Sacha Inchi) and *Caryodendron orinocense* H.Karst. (Cacay) are promising nontimber forest products within Colombia’s Amazonian rainforest, enough to be sustainably exploited. Several authors have previously reported their phytochemical richness—most from the Brazilian Amazon—and recognize some of them as sources of carotenoids, anthocyanins, phytosterols, fatty acids, triglycerides, and diglycerides, and their potential uses in cosmetics and food sectors [1,2,3,6]. The phytochemical analysis of nontimber products allows the identification of biological species that could be used as sources of new products, profiling them to achieve their higher potential, and optimizing the production processes for their industrial use. Thus, bioprospection becomes a business alternative with an increased likelihood of economic success. The oil and fat (this difference depends on its physical state according to ISO 18363-2, 2018) extraction methodologies vary; since this step is the most critical part of the sample processing, different mechanical and chemical techniques have been developed to maintain sample integrity and improve yields. Mechanical and solvent extractions are the most widely used because of their simplicity and relatively low costs; nonetheless, novel techniques such as Enzyme Assisted Extraction (EAE), Ultrasound-Assisted Extraction (UAE), Microwave Assisted Extraction (MAE), and Supercritical Fluid Extraction (SFE) have shown promising results, despite their higher technological requirements [7].

All of the previously named species that are native to the Amazonian region could be used for oil extraction [8,9], so they are potential sources of natural ingredients for a large number of bioproducts. The production of oils and fats in Colombia is critical for different industries, such as manufacturing balanced animal food, soap, toiletries, and pharmaceutical products [10], restricted to African palm, soybean, oilseed oils, and animal fats, considered more profitable. However, nontraditional sources of oils and fats have been undervalued, probably due to the lack of regulation, standardized processes, or the complexity of the matrixes; although, their content of bioactive compounds could be an opportunity to formulate and develop a wider range of bioproducts responding to consumer expectations. It is also necessary to consider several factors which could influence the composition of oils [11], such as the extraction method or plant origin. This work aims to conduct a physicochemical and compositional analysis of oils and fats from native species from the Colombian Amazon region to identify bioactive compounds and gain insight into their potential applications for pharmaceutical, food, and cosmetic (PFC) industries.

## 2. Results and Discussion

The quality indices, physicochemical parameters, and the content of bioactive compounds such as phytosterols, carotenoids, tocopherols, and tocotrienols were evaluated to determine the usage profile of oils/fats extracted from different sources to establish their unique features. Considering that the availability of these oils depends on the harvesting season, quality indices allow us to evaluate the similarities between them and determine whether it is possible to replace one with another in potential applications for PFC industries.

### 2.1. Physicochemical Analysis

Table 1 presents the results of the physicochemical and quality index analysis. According to the results, the highest iodine index corresponds to Cacay, indicating a high content of unsaturated lipids. In the case of Miriti oil, it is similar to that reported by Cardona-Jaramillo et al. [6] (76.4 g I_2_/100 g), which is also of Colombian Amazon origin. We found in Andiroba a value for this index of around 59.2 g I_2_/100 g, which contrasts with the values reported by Araujo-Lima et al. [12] close to 197 g I_2_/100 g for Andiroba from the Brazilian Amazon region. Differences in results for Andiroba may be due to the morphotype of Andiroba and the meteorological conditions of the cultivated area. Even so, lower iodine values have been reported for this species when extraction reaches high temperatures. Silva et al. [13] reported values between 55 and 80 g I_2_/100 g for a sample obtained by drying and expeller pressing that reached temperatures of 90 °C. In contrast, Murumuru butter showed the lowest value, which matches the results of Feitosa et al. [14], who reported Murumuru with the lowest iodine value (11.49 g I_2_/100 g sample) when comparing butter from four Amazonian species from Brazil (Murumuru, Bacurí, Tucumá, and Ucuuba) [15]. Meanwhile, the measured iodine indices of Miriti and Asai oils were lower than that reported by Aquino et al. [16] (90.00 mg I_2_/100 g sample) and by Cardona-Jaramillo et al. [6] (69.2 g I_2_/100 g sample), respectively, indicating that our oils have lower unsaturated lipid content. According to the saponification value, Sacha Inchi, Asai, and Cacay oils have longer chain fatty acids than the other oils evaluated, as they have a saponification index close to 190 mg KOH/g, suggesting the predominant presence of 18-carbon chain fatty acids. On the other hand, Murumuru presents a saponification value of 205 mg KOH/g, indicating a lower average molecular weight of the oil that corresponds to triglycerides of short-chain fatty acids such as lauric (C12) and myristic (C14) [17]. In the end, Andiroba, Asai, and Murumuru showed the highest acidity values suggesting that they presented a fermentative process that allowed enzymatic action related to triacylglycerol hydrolysis and the oxidative process [12]. High acidity index values are an indicator of the poor quality of samples due to the presence of other compounds, such as citric or lactic acid produced by fermentation processes, which modifies the pH and organoleptic properties; for this reason, a low acidity index is a determining factor when a natural ingredient is chosen for the design of cosmetic, pharmaceutical, or food bioproducts. 

In order to compare the six oils under study, the quality indices were used as input variables for a least squares analysis, finding that the samples are distributed in four groups according to the results of the score plot (Figure 1a) and the hierarchical analysis (HCA, available on Appendix A). The statistical distribution of the samples is influenced in the first place by the density based on the loading score (Figure 1b). Likewise, it is established that there is a correlation between acidity, saponification, and refractive indices. Murumuru corresponds to a single group considering that its iodine index is much lower than the other samples. According to the results of the multivariate analysis, it could be inferred that there is a similarity between the group formed by Miriti, Cacay, and Asai. However, their sensory characteristics, such as color, do not allow them to ensure it since their tones are different (orange, yellow, and green, respectively).

### 2.2. Composition Analysis

Given the similarities in some quality indices and physical characteristics of the analyzed oils, we compared the species using the lipid profile, phytosterol, carotenoid, tocopherol, and tocotrienol content since they are molecules of interest in bioproduct formulation.

We found saturated and unsaturated FA chains of 12, 14, 16, and 18 carbon atoms, whose structures have up to 3 double bonds. The oleic acid (C18:1n9c) is predominant in Miriti, Andiroba, and Asai, which also have significant amounts of palmitic acid (C16) (Table 2). The predominance of oleic acid is aligned with the iodine value reported in Table 1 since iodine reacts on double bonds, indicating the presence of unsaturated compounds. However, the content of carotenoids could also result in high iodine values since they possess double bonds. On the other hand, Cacay and Sacha Inchi oils have the highest amount of linolenic acid (C18:3), around 87 and 46%, respectively, followed by linoleic acid (C18:2n6c) with 12.74 and 34.56%.

On the other hand, the Murumuru butter has only saturated fatty acids in its composition, which concurs with the observed iodine index. We found odd-numbered fatty acid chains unusual since they are not common in vegetable oil matrices because of the biochemical pathway involved in their production. However, its presence may be explained by Kroumova et al. [18], and it is also reported in plants by Górnaś et al. and Řezanka and Sigler [19,20]. We hypothesize that its presence could result from endophytic interactions or an artifact associated with sample processing. Other authors related their presence to environmental conditions and the ripening stage [21].

We found omega 9 (oleic) and omega 6 (linoleic) acids in most samples, which are of great importance for their applications in the cosmetic field, where it has been found to help prevent skin dehydration and promote cell membrane regeneration. It also has essential applications in the food industry because its intake helps to control cardiovascular diseases and decrease LDL in the bloodstream [22]. High percentages of monounsaturated and polyunsaturated fatty acids also affect the physical characteristics of oils and fats. Thus, those with a higher rate of unsaturated acids are presented as liquids (Miriti, Cacay, Sacha, and Asai). In contrast, those with saturated acids are found as solids, as in the case of Murumuru. Andiroba is intermediate since it is a viscous liquid oil at 20 °C.

Other compositional analyses were performed and are shown in Table 3; our HPLC methodology displayed high resolution (Chromatograms are available in Appendix A) between peaks in a shorter time than isocratic methods, despite the latter being more often used to analyze these kinds of compounds [23,24]. Miriti presented the highest concentration of α-tocopherol (α-T) of all the oils studied in this paper. Freitas et al. [25] reported a total tocopherol content for this species (α-T and β-tocopherol (β-T)) of 1.041 µg mg^−1^ oil. According to the results, the Miriti oil evaluated in this study has a higher content in these tocols. Other studies reported α-T concentrations of between 0.196 and 1.343 µg mg^−1^ of oil and β-T concentrations of between 0.466 and 0.647 µg mg^−1^ of oil [26]. Cacay and Asai also showed α-T in lower proportions. For Asai oil, alpha-tocopherol (α-T), beta-tocopherol (β-T), gamma-tocopherol (γ-T), and gamma-tocotrienol(γ-T3) were found; however, we were unable to quantify β-T in Asai since it was below the quantification limit (LOQ). In previous research, α-T was found in oil extracted from Asai at concentrations of 0.3219 µg mg^−1^ oil, while γ-T was reported at 0.00796 µg mg^−1^ oil [26,27]. We found β-T and γ-T3 in Andiroba samples, but the latter was below LOQ; this fat is commonly used in the formulation of cosmetic and pharmaceutical products [28]. γ-T and δ-T were the only tocols found in Sacha Inchi oil; however, α-T was not found, even though it was detected in low concentrations (in addition to the two tocopherols already mentioned) in characterizations performed on commercial Sacha Inchi oil by de Souza et al. [29] and Ramos-Escudero et al. [2]. The degree of refinement of the Sacha Inchi oil used in our analysis may affect the results compared to other studies as it is closely related to tocol content. α-T (the widely available form of Vitamin E) is used in food fortification and supplementation. It also plays an essential role in the human body due to its biological activity, including its anticancer, anticholesterolemic, antihypertensive, antioxidant, immunomodulatory, and neuroprotective properties [30,31,32].

Among the unsaponifiable compounds of oils, in addition to tocopherols and tocotrienols, we found phytosterols, which are notable in cosmetics due to their proven photoprotective activity; they act as a barrier against dehydration with the benefit, compared to cholesterol, of a lower probability of suffering from acne [33,34]. At a nutritional level, contrary to what is thought, the consumption of phytosterols helps reduce LDL. These, in turn, are recognized for having anticancer, anti-inflammatory, and antioxidant properties [35]. The widest variety of phytosterols was found in Andiroba, the only sample containing lanosterol and high quantities concerning the others; this is the precursor of the other sterols. The main phytosterol reported is β-sitosterol, found in Murumuru, Sacha Inchi, and Asai oils.

On the other hand, for Miriti, we found stigmasterol as the major one reported for pharmaceutical uses [36]. The presence of carotenoids was reported in Asai and Miriti, the latter having the highest concentration, which was an expected result due to the orange coloration of the oil. In addition to phenolic compounds, tocopherols and carotenoids have a crucial role in the prevention of auto-oxidation and are closely related to the stability of the oil [37].

With all the quantitative data from oil composition analyses, a PCA was performed to determine similarities and relationships among the samples; four groups were found (Figure 2a) where, as in the analysis of quality indices, Murumuru was differentiated, mainly by its lipid profile where saturated short chain fatty acids and high concentration of β-sitosterol prevail. The HCA results are available in Appendix A. We inferred differences for Miriti are given by the high amount of carotenoids, which were negligible in the other samples. Additionally, Miriti oil exhibits high amounts of stigmasterol. For its part, the separation of Asai oil could be attributed not only to its content of carotenoids but also to its stearic and eicosanoic acid content. It is important to note that even if Asai oil showed a content of carotenoids, it seems to be much lower than the concentration of these compounds in Miriti. In terms of composition, Cacay and Sacha Inchi are close regarding lipid profile and phytosterols.

## 3. Materials and Methods

Samples from the Colombian Amazon region were used: Asai oil (*Euterpe precatoria* Mart), Cacay oil (*Caryodendron orinocense* H.Karst.), Murumuru butter (*Astrocaryum Murumuru*), Andiroba oil (*Carapa guianensis*) and refined Sacha Inchi oil (*Plukenetia volubilis*) were obtained from seeds, while Miriti (*Mauritia flexuosa* L.f.) oil was obtained from dry pulp. All oils and fats were obtained mechanically by an expeller press; the conditions were the same as reported by Cardona-Jaramillo et al. [6].

Samples were collected from The El Trueno Experimental Station located in El Trueno hamlet, Retorno Municipality, Guaviare, Colombia (2.373109446700682, -72.63980398357715), between August and December 2019: Sacha Inchi (8/19), Miriti (11/19), Cacay (10/19), Andiroba (12/19), Murumuru (10/19) and Asai (10/19). All samples were stored at 4 °C, protected from light and oxygen to avoid degradation before analysis.

Standards of α, β, γ, and δ-tocopherol; α, β, and γ-tocotrienol were acquired from Sigma-Aldrich. Sodium thiosulphate, boron trifluoride, *N,O*-Bis(trimethylsilyl)trifluoroacetamide with trimethylchlorosilane (BSFTA + TMCS), and pyridine were obtained from Sigma-Aldrich and iodine monobromide from Panreac. All solvents were HPLC grade; hexane, methanol, and ethyl acetate were supplied by Merck and isopropanol (IPA) by Fisher Scientific. Ultrapure water produced by a Milli-Q system was used.

### 3.1. Quality Indices

The acidity index was determined by modifying the protocol proposed by United States Pharmacopeia [38]. A total of 2.5 g of the oil or fat samples was weighed in an Erlenmeyer flask and dissolved in 50 mL of an ethanol-ether mixture in a 1:1 ratio, previously neutralized with 0.01 M sodium hydroxide. The sample was titrated with 0.01 M sodium hydroxide solution using phenolphthalein as an indicator, under heating and stirring. The amount of free fatty acids was calculated and expressed as mg of KOH per gram of sample.

The saponification index was established by modifying the protocol proposed by the United States Pharmacopeia [38]. A total of 2.0 g of the samples was weighed in a 50 mL flask fitted with a ground-glass mouth, and 25 mL of 0.5 N methanolic potassium hydroxide was added and heated under reflux for at least 30 min with constant stirring. Finally, phenolphthalein was added, and the excess potassium hydroxide was titrated with 0.5 N hydrochloric acid. The saponification number is expressed in terms of mg of KOH required to saponify one gram of the substance.

For the iodine value, 1.0 g of the oil or fat was weighed in an Erlenmeyer flask, 10 mL of chloroform and 25.0 mL of iodine bromide solution (2 g/L) were added. The container was covered and left to stand without light for 30 min. Then, 30.0 mL of potassium iodide and 100 mL of distilled water were added. The liberated iodine was titrated with 0.1 N sodium thiosulfate until the yellow coloration vanished. Then, 3.0 mL of saturated starch solution was added and titrated until the blue color disappeared. The iodine value was defined as the grams of iodine absorbed per 100 g of substance [38].

### 3.2. Fatty Acid Profile

Derivatization was conducted using the UNE-EN ISO 661:2006 methodology to obtain the fatty acid profiles [39]. A total of 50 mg of each sample was dissolved in 6 mL of fresh 0.5 M methanolic sodium hydroxide solution. The solution was heated under reflux until the disappearance of the fat droplets was observed (30–60 min). Then, 7.0 mL of methanolic boron trifluoride (BF_3_) solution was added, and the mixture boiled for three more minutes. Next, 3.0 mL of isooctane was added to the top of the condenser, and the flask was removed from the heating plate. While hot, 20.0 mL of saturated brine was added, and the covered flask was vortexed for 15 s. Then, another 10.0 mL was added to generate a phase separation. A 2.0 mL aliquot of the organic phase was taken and dried with 0.2 g of anhydrous sodium sulfate. Finally, a 100 µL aliquot of the supernatant was taken and brought to 1.0 mL with isooctane. Analysis was performed on an Agilent 7890B Gas Chromatograph coupled to an FID detector, using an Agilent DB23 column (60 m × 0.25 µm × 250 µm), Helium Grade 5.0 stripping gas with a 1.1 mL/min (constant) flow, the temperature of the injector was 270 °C in split mode (30:1). The initial oven temperature was 60 °C (1 min), 6 °C/min up to 210 °C (24 min), detector temperature was 310 °C, air flow of 400 mL/min, hydrogen flow 40 mL/min, makeup gas (He) at 30 mL/min (compensated with column flow), and a solvent delay of 9 min. Fatty acid analysis was conducted using a certified standard (SUPELCO, Supelco 37 Component FAME Mix) and using the external standard method; additionally, a GC-MS (Agilent 5977A MSD) analysis was performed to ensure the identification of the derivatized fatty acids using the NIST14 library. Finally, retention indices were calculated using n-Paraffin mix standard C10-C24 (Sigma-Aldrich, St. Louis, MO, USA), and the results were compared with the literature (Retention indices are available in Appendix A).

### 3.3. Phytosterol Analysis

The phytosterol analysis was conducted with the unsaponifiable fraction, starting from 2.5 g of fat, adding 25 mL of 12% ethanolic KOH solution, and heated under reflux for two hours. It was transferred to a separatory funnel, 25 mL of distilled water was added, the mixture was shaken, and two liquid−liquid extractions were carried out with 10 mL of petroleum ether; the organic phase was dried using anhydrous sodium sulfate and filtered. Finally, it was concentrated under reduced pressure and quantitatively transferred to a previously weighed vial [40,41]. Samples were prepared using 1.0 mg of the unsaponifiable fraction, and a solution of 1000 ppm of cholesterol standard was used as a reference. Then, 100 µL of BSFTA+TMCS and 100 µL of pyridine were added to the dry samples. The mixture was homogenized and heated at 80 °C for 30 min. Analysis was performed on an Agilent 7890B Gas Chromatograph coupled to an FID detector, using an Agilent HP5-MS column (30 m × 0.25 µm × 250 µm), Carrier gas Helium Grade 5.0 at 1 mL/min (constant) flow, injector at 270 °C in Split mode (15:1), Initial oven temperature 250 °C (1 min), 2 °C/min up to 300 °C (7 min), Detector Temperature 300 °C, Air Flow 400 mL/min, Hydrogen Flow 40 mL/min, Makeup (He) 30 mL/min, and a solvent delay of 9 min [42].

### 3.4. Tocopherol and Tocotrienol Analysis

Tocopherol and tocotrienol were analyzed using an Agilent 1260 Infinity II UHPLC instrument equipped with an autosampler and a quaternary pump, coupled to an Agilent G7121A 1269 FLD fluorescence detector (Agilent, Santa Clara, CA, USA). The Luna Silica column (250 × 4.6 mm, 3 µm) (Phenomenex, Torrance, CA, USA) was used, and a binary mobile phase of n-hexane (A) and ethyl acetate (B) in gradient (0–2 min 100%A, 2–5 min 98%A, 5–14 min 93%A, 14–15 100%A, 15–18 min 100%A), with an injection volume of 10 µL, a constant flow rate of 1.7 mL min^−1^ and a total time of 18 min, with an oven temperature of 40 °C. Fluorescence detection was performed with excitation and emission wavelengths of 297 nm and 328 nm, respectively [43,44]. The identification and quantification were conducted using α, β, γ, and δ-tocopherol; α, β, and γ-tocotrienol standards (Sigma-Aldrich, St. Louis, MO, USA).

### 3.5. Carotenoid Analysis

Total carotenoid content was performed using the spectrophotometer GENESYS 10S UV-Vis (ThermoScientific, Waltham, MA, USA). A total of 1.5 mL of n-hexane was added to 100 mg of the sample and mixed in a vortex for 1 h. The samples were centrifuged, and the absorbance was measured at 450 nm. The total carotenoid content was calculated using the formula used by De Carvalho [45].

### 3.6. Statistical Analysis

All experiments were performed in triplicate for each sample and the results were expressed as mean and standard deviation (SD). Since we are comparing oils from different species, a multivariate analysis was conducted. We compared the samples using principal component analysis (PCA) and hierarchical cluster analysis (HCA) with input variables such as density, refractive index, and quality indices (acidity, saponification, and iodine value). Additionally, another PCA and HCA were performed with another dataset to identify clusters using input variables of absolute concentrations from compositional analysis (carotenoid content, phytosterol content, tocol content) and the relative percentage of fatty acids. All analyses were performed using SIMCA software (version 14.1, Umetrics).

## 4. Conclusions

Results displayed that analyzed samples could be classified into four groups, guided mainly by their chemical composition rather than physical properties. Although the quality indices are a valuable tool for evaluating a single sample of oil/fat material, they are not the most suitable choice for comparison. The multivariate analysis exhibits a unique composition for every species, with properties so different that the oils/fats cannot be exchangeable. The first group was Miriti; this oil was the most promising raw material for the cosmetic industry due to its high content of unsaturated fatty acids (ω-9), tocopherols, and phytosterols. Its high unsaturation grade, explained by its high iodine equivalent, would make it susceptible to oxidation processes; however, the high concentration of antioxidants, such as tocopherols and carotenoids, gives it higher stability and resistance. Murumuru’s group was mainly composed of saturated fatty acids and β-sitosterol, which made it more suitable for the cosmetic rather than the food industry. The third group corresponded to Asai, composed exclusively of a mixture of saturated and unsaturated fatty acids (palmitic and oleic) and a significant amount of β-sitosterol, making it a versatile product for functional foods or dermocosmetics. Finally, the Cacay, Sacha Inchi, and Andiroba group reported a more complex composition with a lower concentration of various metabolites but a high concentration of unsaturated fatty acids like oleic, linoleic, and linolenic acids that makes them susceptible to oxidation reactions; therefore, their storage is a challenge.

In conclusion, all the studied species possessed a vast amount of lipids vital for PFC industries. They all contain a significant quantity of bioactive metabolites essential for manufacturing high-value products. These species have become key players in producing high-quality natural ingredients. These results showed a promising scenario for strengthening the value chain of Amazon nontimber forest species as an alternative for the region’s development.

## Figures and Tables

**Figure 1 biomolecules-13-00985-f001:**
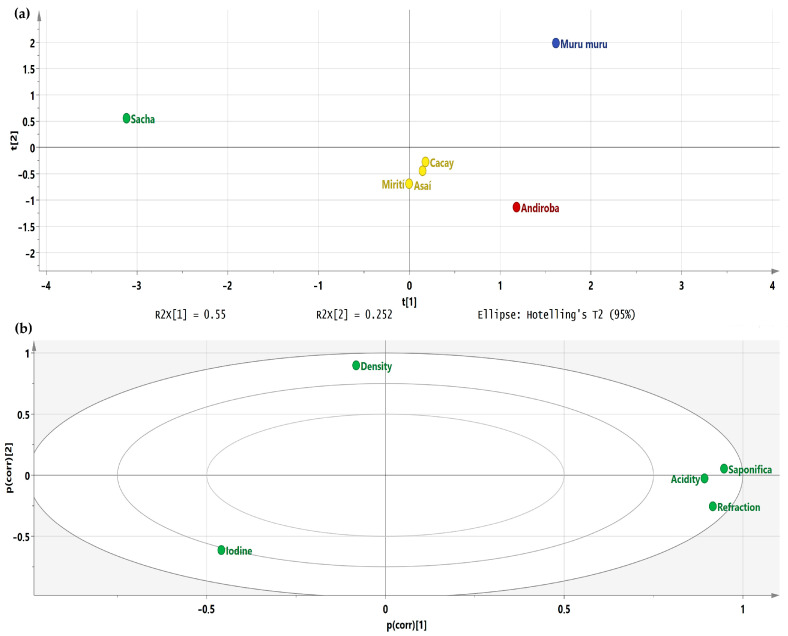
(**a**) Score plot based on the quality indices and physicochemical properties. This analysis explains more than 80% of the total variance of the data used. The *x*-axis (T[1]) explains 55% of the total variance, the *y*-axis (T[2]) explains 25.2% of the total variance; (**b**) the Loading plot is based on the quality indices and physicochemical properties. The *x* and *y* axes are Pearson correlation coefficients.

**Figure 2 biomolecules-13-00985-f002:**
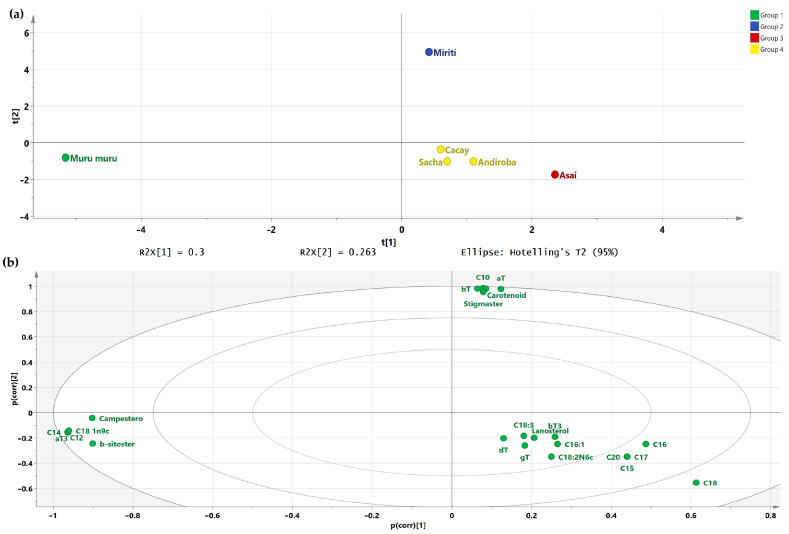
(**a**) Score plot based on composition analysis. This analysis explains more than 55% of the total variance of the data used. The *x*-axis (T[1]) explains 30% of the total variance. The *y*-axis (T[2]) explains 26.3% of the total variance; (**b**) Loading plot based on composition analysis. The *x* and *y* axes are Pearson correlation coefficients.

**Table 1 biomolecules-13-00985-t001:** Quality indices and properties of oils and fats of Amazonian origin.

Oil Source	Oil Yield/Content (% *w/w)*	Saponification (mg KOH/g Sample)	Iodine (g I_2_/100 g Sample)	Acidity (mg KOH/g Sample)	Density (g/mL)	Refractive Index
Miriti	66.63	189.21 ± 0,70	76.38 ± 0.70	3.68 ± 0.70	0.9133 ± 0.0002	1.4656 ± 0.0003
Cacay	69.57	166.42 ± 1.69	253.63 ± 5.88	3.61 ± 0.36	0.9061 ± 0.0004	1.4711 ± 0.0003
Murumuru	71.33	205.1 ± 4.70	19.10 ± 0.47	7.28 ± 0.11	0.9264 ± 0.0001	1.4504 ± 0.0001
Andiroba	73.68	189.36 ± 3.56	59.2 ± 2.98	8.31 ± 0.11	0.9054 ± 0.0001	1.4663 ± 0.0002
Sacha Inchi	76.23	123.10 ± 1.86	65.20 ± 0.41	1.00 ± 0.02	0.9184 ± 0.0002	ND ^1^
Asai	55.55	170.40 ± 9.01	63.40 ± 0.43	5.51 ± 0.01	0.9133 ± 0.0002	1.4687 ± 0.0003

^1^ Not determinable.

**Table 2 biomolecules-13-00985-t002:** Fatty acid methyl ester (FAME) profiles of oils and fats from Amazonian origin.

	Lipidic Profile (Relative %)						
Fatty Acid		Miriti	Cacay	Murumuru	Andiroba	Sacha Inchi	Asai
Capric	C:10	0.01 ± 0.00	-	-	-	-	-
Lauric	C:12	0.08 ± 0.00	-	64.26 ± 1.29	-	-	-
Myristic	C:14	0.18 ± 0.00	-	26.57 ± 0.53	-	-	0.1 ± 0.00
Pentadecylic	C:15	-	-	-	-	-	0.1 ± 0.00
Palmitic	C:16	6.69 ± 0.13	0.33 ± 0.00	3.39 ± 0.06	20.2 ± 0.40	4.96 ± 0.10	16.7 ± 0.33
Palmitoleic	C16:1	-	-	-	0.8 ± 0.016	-	0.1 ± 0.00
Margaric	C:17	-	-	-	-	-	0.1 ± 0.00
Stearic	C:18	-	0.16 ± 0.00	-	5.3 ± 0.11	2.99 ± 0.06	6.8 ± 0.14
Oleic	C18:1n9c	93.04 ± 1.86	-	3.19 ± 0.06	45.5 ± 0.91	11.1 ± 0.22	70 ± 1.4
Linoleic	C18:2n6c	-	12.74 ± 0.26	1.22 ± 0.02	8.3 ± 0.17	34.56 ± 0.69	3.3 ± 0.07
Arachidic	C:20	-	-	-	-	-	0.4 ± 0.01
Linolenic	C18:3	-	86.77 ± 1.74	-	0.8 ± 0.02	46.39 ± 0.93	0.8 ± 0.02

**Table 3 biomolecules-13-00985-t003:** Carotenoid, phytosterol, tocopherol, and tocotrienol composition of oils and fats of Amazonian origin.

	Tocopherols, Tocotrienols (µg mg^−1^ Sample)						Phytosterols (µg mg^−1^ Sample)				Carotenoids (mg 100 g^−1^ Sample)
	αT	αT3	βT	γT	γT3	δT	Stigmasterol	Campesterol	β-Sitosterol	Lanosterol	
Miriti	1.168 ± 0.008	-	1.608 ± 0.017	-	-	-	3.824 ± 0.129	0.126 ± 0.029	0.500 ± 0.006	-	128.19 ± 8.14
Cacay	0.0413 ± 0.002	-	<LOQ	0.4436 ± 0.011	-	-	-	-	-	-	0.12 ± 0.00
Murumuru	-	0.034 ± 0.002	0.057 ± 0.002	-	0.021 ± 0.005	-	0.245 ± 0.005	0.636 ± 0.013	5.257 ± 0.503	-	-
Andiroba	-	-	0.161 ± 0.008	-	<LOQ	-	0.203 ± 0.019	0.248 ± 0.006	0.679 ± 0.056	2.312 ± 0.123	-
Sacha Inchi	-	-	-	1.795 ± 0.042		0.795 ± 0.046	0.937± 0.020	-	2.265 ± 0.123	-	-
Asai	0.085 ± 0.005	-	<LOQ	0.1715 ± 0.001	0.222 ± 0.015	-	0.141 ± 0.015	-	1.458 ± 0.123	-	2.32 ± 0.29

## Data Availability

The data presented in this study are available in the article.

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
