# Peer review of "Comprehensive Characterization of Oils and Fats of Six Species from the Colombian Amazon Region with Industrial Potential"

_biomolecules, 2023, doi:10.3390/biom13060985_

Round 1

Reviewer 1 Report

The manuscript from Lozano-Garzon reports the physical properties and fatty acid content of oils extracted from five non woody species native to Colombia's Amazon rainforest. This is of interest because of the commercial potential of these species. The work follows standard protocols and is clearly reported. Some minor editing of the English might help some of the phrasing.

At the start of the results section it would help the reader if details were added information about how the oil samples were obtained. What part of the plants was used, and what sort of process was used to isolate the oils.

In Table 1 please add some units for oil yield/content. Was this by weight, ie g/g, or something else?

In Table 1 the "Cacay" product has a low Iodine index, an indication of a low content of unsaturated fatty acids, but in Table 2 the major fatty acid of this sample is an unsaturated one. Please explain how these results, that appear to suggest different conclusions, can be reconciled.

In the Methods section please describe where samples were collected and at what time of year.

The English is generally acceptable. There are a few occurrences where the phrasing is convoluted or an expression not used in English appears to have been translated. Minor editing by a native English speaker should fix this.

Author Response

Dear reviewer

Thank you for allowing us to submit a revised draft of our manuscript "Comprehensive characterization of oils and fats of six species from the Colombian Amazon region with industrial potential" to Biomolecules. We appreciate the time you dedicated to providing us feedback on our manuscript. In general terms, we adjusted some aspects of the manuscript addressing all your suggestions, using track control for all of them to make it easier to follow the manuscript's changes. You will find in the document attached the responses to all your questions and suggestions.

We look forward to the next stage of the review process.

Reviewer 2 Report

The compound identification and retention indices are more important factor for fatty acids profiling. I would like author to add those. 

Good

Author Response

(The authors gave the same response as above.)

Reviewer 3 Report

The study deals with the characterisation of oils and fats of six different species in the Amazon region with industrial potential. The results of this study are interesting for different stakeholders, but the topic "industrial potential" was not addressed very much. Furthermore, the research task was only rudimentarily defined. What was the challenge (e.g. analytical, mathematical or technological) in this study? What does this study distinguish it from others?

The PCA method used is not a cluster analysis (see line 130 - hierarchical analysis (HCA) - Hierarchical Cluster Analysis). A PCA method reduces the variables, a cluster analysis finds the same. It was mentioned that the SIMC method was used. However, in the results part only the PCA method was mentioned. What was really used? How many data sets were used?

The HPLC chromatograms are helpful, so I would recommend that these are included in the manuscript.

The extraction method is not traceable (line 244: “The extraction details are not shown since those are part of an intellectual property saved process”), please can you describe the process here? If the process is to be kept secret, then the authors should consider that the manuscript should not submitting for publication.

What databases were used to analyse the GC and UHPLC data?

What is the industrial potential of the oils and fats analysed now? Please describe this briefly in the conclusions.

Author Response

(The authors gave the same response as above.)

Round 2

Reviewer 1 Report

Thank you for making corrections to the manuscript. I think it is much improved and now ready for publication

Author Response

Dear reviewer,

Thank you very much for your helpful comments and suggestions.

Reviewer 3 Report

The authors reworked and improved the manuscript. However, not all points for the publication have been fulfilled yet. On the one hand, the research question and the aim of the study were only rudimentarily presented. Secondly, the statistical investigation is not sufficiently described. The results of the HCA were presented, but not described in the method section. Furthermore, it is clearly defined which variables (e.g. GC-Chromatogram, relative concentrations of the substances, …) were used for the respective studies (PCA, HCA).

Author Response

Dear reviewer

We appreciate the time you dedicated to providing us with feedback on our manuscript. In general terms, we adjusted the Methods section addressing all your suggestions and enhanced the aim of our study.  You will find in the document attached the responses to all your suggestions.

We look forward to the next stage of the review process.
